# Enantiopure trigonal bipyramidal coordination cages templated by in situ self-organized $D_{2h}$-symmetric anions

Shan Guo[1,3], Wen-Wen Zhan[1,3], Feng-Lei Yang[1,3], Jie Zhou[1], Yu-Hao Duan[1], Dawei Zhang [2] ✉ & Yang Yang [1] ✉

The control of a molecule's geometry, chirality, and physical properties has long been a challenging pursuit. Our study introduces a dependable method for assembling $D_3$-symmetric trigonal bipyramidal coordination cages. Specifically, $D_{2h}$-symmetric anions, like oxalate and chloranilic anions, self-organize around a metal ion to form chiral-at-metal anionic complexes, which template the formation of $D_3$-symmetric trigonal bipyramidal coordination cages. The chirality of the trigonal bipyramid is determined by the point chirality of chiral amines used in forming the ligands. Additionally, these cages exhibit chiral selectivity for the included chiral-at-metal anionic template. Our method is broadly applicable to various ligand systems, enabling the construction of larger cages when larger $D_{2h}$-symmetric anions, like chloranilic anions, are employed. Furthermore, we successfully produce enantiopure trigonal bipyramidal cages with anthracene-containing backbones using this approach, which would be otherwise infeasible. These cages exhibit circularly polarized luminescence, which is modulable through the reversible photo-oxygenation of the anthracenes.

Template synthesis is an efficient way to access complex structures[1-4]. Anions play a significant role as established templates in the coordination-driven self-assembly of metal-organic macrocycles and cages, thereby enhancing yields and producing cleaner products in dynamic combinatorial libraries[5-8]. The resulting structures directed by anions are strongly influenced by the shapes, charges, and sizes of the anion templates, which typically occupy the cavities within the metal-organic assemblies[9-11]. Sun and coworkers demonstrated a series of anions could induce multiple metal-organic macrocycles formed from the same building blocks[12], and tetrahedral anions were able to template helicates aggregating into a tertiary-like hexamer structure[13]. Fujita and coworkers exhibited anion exchanges triggered reconfigurations of a class of entangled metal-organic assembly[14,15]. The structural outputs also serve as a reflection of the complexity inherent in anion templates. The Nitschke's group has reported a range of interesting structures templated by the synergistic effects of paired anions[16] and two distinct types of anions[17,18]. However, previous studies of the template effects have mainly focused on pristine simple anions. In certain biological systems, proteins are known to self-assemble initially and then act as templates for subsequent aggregations[19,20]. The utilization of in situ self-organized anions as templates in metal-organic assembly is thus more appealing and challenging[21].

Coordination cages, composed of organic ligands and metal ions that self-assemble to form three-dimensional discrete frameworks with internal cavities[22-25], have garnered significant attention for their broad applications in sensing[26,27], separations[28-30], stabilization of labile species[31-33], and catalysis[34-37]. Extensive efforts have been dedicated to the rational design of these self-assembled architectures to enhance

[1]School of Chemistry and Materials Science, Jiangsu Normal University, Xuzhou 221116, China. [2]State Key Laboratory of Petroleum Molecular & Process Engineering, Shanghai Key Laboratory of Green Chemistry and Chemical Processes, School of Chemistry and Molecular Engineering, East China Normal University, Shanghai 200062, China. [3]These authors contributed equally: Shan Guo, Wen-Wen Zhan, Feng-Lei Yang. ✉e-mail: dwzhang@chem.ecnu.edu.cn; yangyang@jsnu.edu.cn

their structural and functional complexity. Leveraging symmetry considerations, strategic ligand design, and elaborative selection of metal ions enable the creation of diverse polyhedral shapes including tetrahedra, octahedra, cubes, and cuboctahedra[38–40]. These polyhedra typically possess high symmetry. However, accessing the less symmetric trigonal bipyramidal cages ($D_3$ symmetry) in a predictable way remains challenging[41], despite a few reported instances[42–45]. Chirality is a crucial aspect in natural systems[46]. Chiral cages create specialized chiral environments within their cavities, facilitating chiral sensing, chiral separation, and asymmetric catalysis[47–51]. Notably, enantiopure trigonal bipyramidal cages are relatively uncommon compared to other highly symmetric polyhedra, primarily due to the inherent difficulty in simultaneously controlling both geometry and chirality. Moreover, chiral selectivity is commonly observed in chiral cages when interacting with small chiral organic molecules[52]. In contrast, the enantioselective incorporation of chiral-at-metal complexes into homochiral cages is a rare phenomenon[53].

In this work, based on our previous studies on chiral cages[54–56], we describe a unique strategy for the construction of chiral trigonal bipyramidal cage structures utilizing the template effect of chiral-at-metal anions, which are formed by the self-organization of $D_{2h}$-symmetric anions around a metal ion. Interestingly, these $D_{2h}$-symmetric anions can induce a structural transformation from a tetrahedral to a trigonal bipyramidal geometry with additional metal ions. Furthermore, the resulting trigonal bipyramidal cages exhibit chiral selectivity for the incorporated chiral-at-metal anion complexes, whose chirality is dictated by the point chirality of the chiral amines applied in forming the ligands. The proposed methodology is universally applicable to various ligands and allows for the assembly of larger cages with enlarged $D_{2h}$-symmetric anions as templates. Importantly, we successfully fabricate a trigonal bipyramidal cage with anthracenes in the backbones using this approach that would otherwise be inaccessible. The resultant cages exhibit circularly polarized luminescence (CPL) that can be modulated by the reversible photo-oxygenation of the anthracene units.

## Results

### Template effect of in situ self-organized oxalate anions ($ox^{2-}$)

Rigid linear $C_2$-symmetric ligands reacting with octahedral transition metal ions usually result in $M_4L_6$ tetrahedra[57,58]. As shown in Fig. 1a, after refluxing **A** with amines to form bis-pyridylimine ligands (**L$^1$**) and then incorporating them with zinc salts (BF$_4^-$, ClO$_4^-$ or NTf$_2^-$ = bis-(trifluoromethylsulfonyl)imide), a [Zn$_4$(**L$^1$**)$_6$]$^{8+}$ cage (**1**) was formed, which was verified by electrospray ionization mass spectrometry (ESI-MS, Supplementary Fig. 10). The $^1$H NMR spectrum of **1** (Fig. 1e) indicated the formation of a high-symmetry product in solution, with only half of the proton signals of the ligand being observed. Single crystal X-ray diffraction (SCXRD) revealed that $R$-**1** has a $T$-symmetric tetrahedral geometry (Fig. 1b). Similar to previous reports[59], vertexes of $R$-**1** consist of Zn-tris(pyridylimine) units in a $fac$-$\Delta$ configuration, and the handedness relates to the stereo-configuration of the chiral amine. The chiral information is transferred through the rigid ligands. The phenyl groups near the stereogenic center communicate with the pyridyl ring of adjacent ligands via face-to-face π···π interactions. The CD studies (Fig. 1h) indicate all the cages in the crystal are homochiral.

When K$_3$[Cr(ox)$_3$] was added as a template into the reaction mixture (Fig. 1a), a trigonal bipyramidal structure, **2-Cr**, was obtained. The ESI-MS study of the acetonitrile solution of the samples showed a clean spectrum with a series of multi-charged peaks (+3 and +4), as shown in Fig. 1f. These peaks corresponded to [Cr$^{III}$(ox)$_3$⊂Zn$^{II}$$_5$(**L$^1$**$_6$)] (NTf$_2$)$_{7-n}$]$^{n+}$ ($n$ = 3 and 4). $R$-**2-Cr** crystallized in the chiral $R_3$ space group, adopting a [Cr(ox)$_3$]⊂Zn$_5$L$_6$ trigonal bipyramidal structure, as depicted in Fig. 1c. The chiral-at-metal $D_3$-symmetric [Cr(ox)$_3$]$^{3-}$, located at the center, templates the formation of such structure. The distance of Zn$^{II}$ – Zn$^{II}$ from two poles is approximately 23.3 Å, while

the edge lengths of the equatorial Zn$^{II}$$_3$ triangle are around 9.0 Å. The apical vertexes consist of Zn-tris(pyridylimine) units in a $fac$-$\Lambda$ configuration. The waist region of the structure is composed of three Zn-bis(pyridylimine) units in a $N_{py},N_{py}$-$trans$-$\Lambda$ configuration (Fig. 1d). The equatorial Zn$^{II}$ atoms have two remaining coordination sites that are occupied by oxygen atoms from one of the ox$^{2-}$ ligands of the chromium(III)-trisoxalate complex. This complex, in a $\Delta$ configuration, occupies the center of the structure and divides the framework into a twin-cage.

The structure of [Cr(ox)$_3$]$^{3-}$ exhibits opposite handedness compared to the Zn$_5$(**L$^1$**)$_6$ framework (Fig. 1d), resembling other documented 2-dimensional structures based on chromium(III)-trisoxalate complexes[60,61]. Therefore, the overall configuration of the structure can be described as $\Delta$-[Cr$^{III}$(ox)$_3$⊂$\Lambda_5$-$R$-Zn$_5$(**L$^1$**)$_6$. The point chirality of the $R$−1-phenylethylamine defined the Zn$_5$L$_6$ framework to be a $\Lambda_5$ configuration, which enantio-selectively encloses a chiral-at-metal $\Delta$-[Cr(ox)$_3$]$^{3-}$ complex. This conclusion is further supported by circular dichroism (CD) studies (Fig. 1h). Interestingly, when one equivalent of racemic K$_3$[Cr$^{III}$(ox)$_3$] is added to a solution containing 6 equivalents of **L$^1$** and 5 equivalents of Zn salts, an isolation yield of approximately 70% for **2-Cr** is achieved (Supplementary Fig. 11). This is noteworthy because racemic K$_3$[Cr$^{III}$(ox)$_3$] contains only 0.5 equivalent of the $\Delta$ configuration. Since K$_3$[Cr$^{III}$(ox)$_3$] can undergo racemization at room temperature[62], it is inferred that the template formation process drives the other 0.5 equivalent of $\Lambda$ configuration into its antipode. This highlights the significant influence of the chirality of sub-components on the handedness of the templates, while the template determines the trigonal bipyramidal arrangement of the sub-components.

The paramagnetic chromium ion hinders the study by $^1$H NMR and [Cr$^{III}$(ox)$_3$]$^{2-}$ is a pre-organized chiral-at-metal complex. Therefore, we attempted to directly use potassium oxalates as templates instead of the pre-formed K$_3$[Cr$^{III}$(ox)$_3$]. Interestingly, this resulted in the formation of yellow crystalline products **2-Zn** with the formula {[Zn$^{II}$(ox)$_3$⊂Zn$^{II}$$_5$(**L$^1$**)$_6$}·6BF$_4$ in high yield (92% as isolated crystals). The composition was verified by ESI-MS analysis (Supplementary Fig. 16). X-ray photoelectron spectroscopy experiments also supported the exclusive presence of zinc as metal in the crystals (Supplementary Fig. 17). This indicates that the $D_{2h}$-symmetric oxalates undergo in situ self-organization into a chiral-at-metal complex [Zn$^{II}$(ox)$_3$]$^{4-}$, which then acts as a template for the formation of **2-Zn**. In the $^1$H NMR spectrum of **2-Zn** (Fig. 1e), twice as many proton signals were observed compared to **1**, indicating a lower symmetry. The two ends of the linear $C_2$-symmetric ligand were no longer in identical environments, which is consistent with the $D_3$-symmetry of the crystal structure of **2-Cr**. Therefore, **2-Zn** shares an iso-structure with **2-Cr**. Interestingly, the addition of 3 equivalents of potassium oxalates and 2 equivalents of Zn salts into the solution of **1** transformed it into **2-Zn** (Fig. 1e and Supplementary Fig. 18). This experiment indicates that the self-organized chiral-at-metal complex [Zn$^{II}$(ox)$_3$]$^{4-}$ was able to template the formation of **2-Zn**, despite the other components being an intact entirety. Moreover, **2-Zn** is more thermodynamically favorable than **1** when oxalate ions are present in the solution. Additionally, we attempted to titrate potassium oxalates into solution of **1** until excess without additional Zn salts, resulting in the decomposition of **1** into free ligands (Supplementary Fig. 19). These observations suggest that the oxalate ion is a stronger ligand than the pyridylimine. The introduction of oxalate ions promoted the formation of coordination bonds between Zn$^{II}$ ions and oxalate ions, disrupting the coordination bonds between pyridylimine and Zn$^{II}$ ions, potentially serving as a driving force for the transformation. ESI-MS of the transforming mixture suggested [(**L$^1$**)$_2$Zn$_4$(ox)$_2$(BF$_4$)$_2$(H$_2$O)$_1$]$^{2+}$ may be a potential intermediate (Supplementary Fig. 20).

To confirm the importance of the anionic feature of ox$^{2-}$ in the self-organized template effect, we substituted 2,2′-bipyrimidine (bpm) for ox$^{2-}$ in the identical synthetic process to that of **2-Zn**. Bpm shares

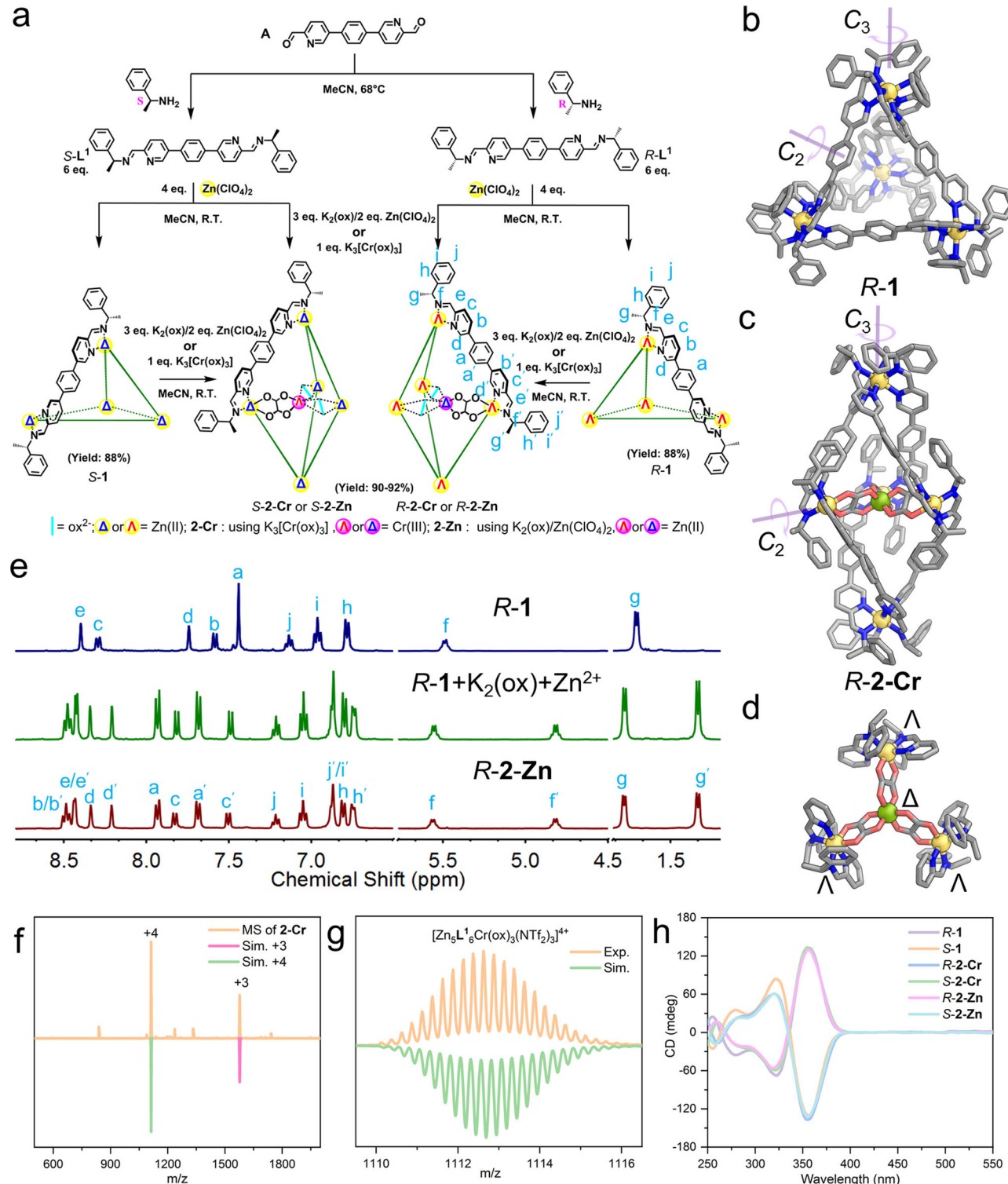

**Fig. 1 | Template synthesis of cages by self-organized ox²⁻ anions. a** The preparation of cage **1**, **2-Cr** and **2-Zn**. X-ray crystal structures of cation parts of (**b**) *R*-**1** and (**c**) *R*-**2-Cr**. **d** The handednesses of the equatorial metal centers of *R*-**2-Cr** with chiral-at-metal complex [Cr$^{III}$(ox)₃]³⁻ as a template in the center; color codes: C, gray; N, blue; O, red; Zn, yellow; Cr, yellow-green. Hydrogens were omitted for clarity. **e** ¹H NMR spectra (CD₃CN, 400 M, 298 K) of *R*-**1**(blue), reaction product of

mixing *R*-**1** with K₂(ox) and Zn salts (green), and *R*-**2-Zn** (red). **f** ESI-MS spectrum of **2-Cr** with molecular peaks being labeled and compared with simulated peaks (downward ones). **g** Experimental and simulated isotopic patterns of 4+ charged molecular peak of **2-Cr**. **h** CD spectra of *R/S*-**1** (0.011 μM), *R/S*-**2-Cr** (0.013 μM) and *R/S*-**2-Zn** (0.013 μM) in acetonitrile.

similarities with the ox²⁻ ion, such as $D_{2h}$-symmetry, bidentate chelating ability on both sides, and similar separations of coordination atoms. However, the major difference is that bpm is a neutral ligand. In the reaction mixture, only **1** was observed in the ¹H NMR spectrum

(Supplementary Fig. 22), indicating that 2,2′-bipyrimidine was unable to self-organize into a chiral-at-metal template to induce the formation of a trigonal bipyramid. Furthermore, [Ru(bpm)₃]²⁺, a $D_3$-symmetric cation with pre-organized bpms, also failed to induce the assembly of a

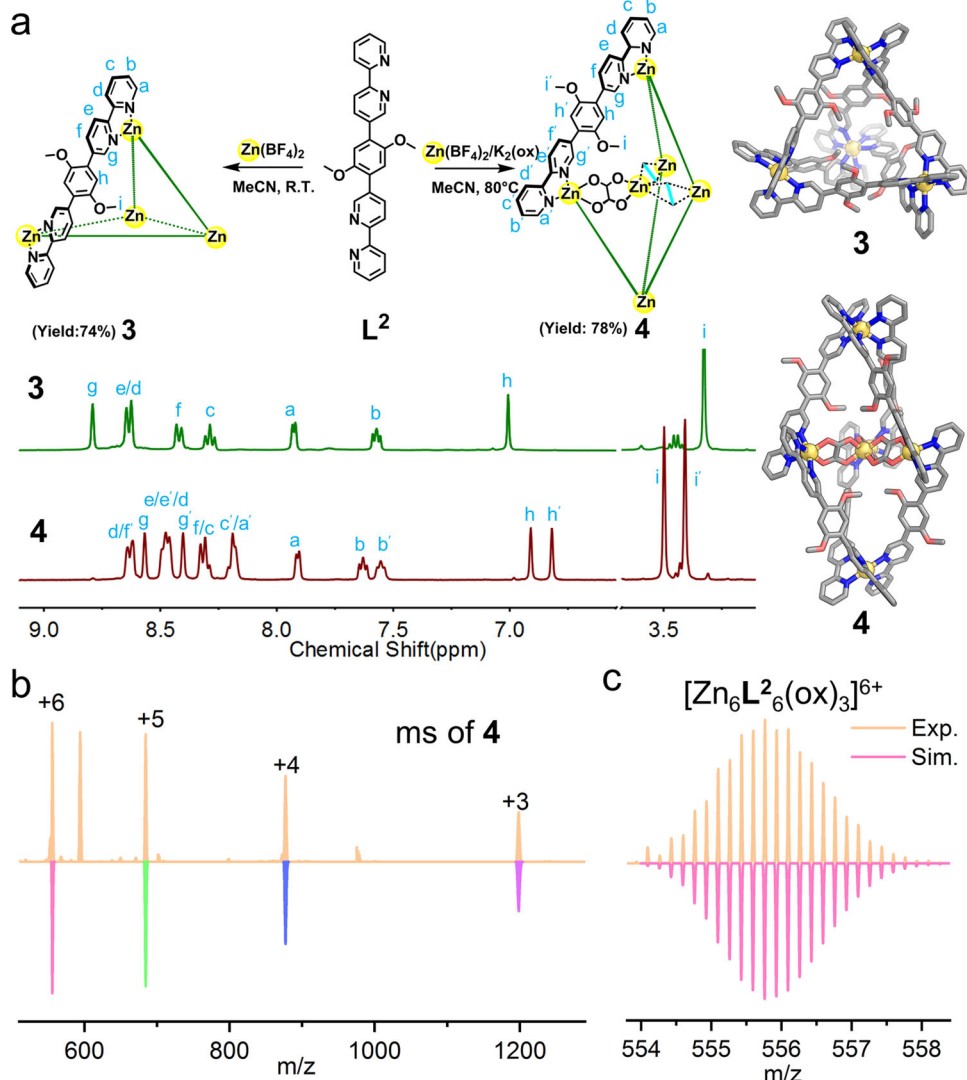

**Fig. 2 | Template synthesis with ligands of different binding node. a** The preparation of cage **3** and **4**, DFT optimized structures of **3** and **4** on B3LYP/genecp (6−31 G(d) for C, H, N, O, lanl2dz for Zn) level (color codes: C, gray; N, blue; O, red; Zn, yellow; H, omitted), and $^1$H NMR spectra (CD$_3$CN, 400 M, 298 K) of **3** (green) and **4** (red). **b** ESI-MS spectrum of **4** with molecular peaks being labeled and compared with simulated peaks (downward ones). **c** Experimental and simulated isotopic patterns of 6+ charged molecular peak of **4**.

trigonal bipyramidal cage (Supplementary Fig. 22). These experiments suggest that the anionic feature of the templates is a key factor in the formation of such a cationic cage.

## Efficiency of the in situ self-organized templates for other ligands

We also designed ligand **L²** (Fig. 2a) with 4,4'-bipyridyl moieties as binding units to test the efficiency of the self-organized template. As expected, when in the absence of the template during self-assembly, the ESI-MS analysis suggested the formation of a [Zn$_4$(**L²**)$_6$]·8BF$_4$ (**3**) cage (Supplementary Fig. 26). The $^1$H NMR spectrum of **3** demonstrated (Fig. 2a) a half set of the ligand signals corresponding a high-symmetry structure. In contrast, when potassium oxalate was added, the resulting product **4** exhibited a complete set of proton signals of the $C_2$-symmetric ligand in the $^1$H NMR spectrum, as shown in Fig. 2a, in accordance with a $D_3$-symmetric structure similar to **2-Zn**. The ESI-MS spectrum of **4** suggested a {[Zn$^{II}$(ox)$_3$]⊂Zn$^{II}_5$(**L²**)$_6$}·6BF$_4$ formula, as shown in Fig. 2b, c. Consequently, **4** also has a trigonal bipyramidal structure with the inclusion of the self-organized [Zn$^{II}$(ox)$_3$]$^{4-}$ as templates. The density functional theory (DTF) optimized structures of **3** and **4** are shown in Fig. 2a (Supplementary Data 1). These experiments

verified that the synthetic protocol of synthesizing trigonal bipyramidal cages works efficiently for other ligands having different chelating moieties.

The reliable template effect of the self-organized $D_{2h}$-symmetric anions encouraged us to explore other similar anions as templates. We hypothesized that a larger $D_{2h}$-symmetric anion resembling ox$^{2-}$ would be able to construct larger trigonal bipyramidal cages. Chloranilic anion (chl$^{2-}$) is an ideal candidate (Fig. 3a), which illustrates an enlarged version of the oxalate anion, sharing the same −2 charge, identical $D_{2h}$-symmetry, and bidentate chelating ability. However, the reaction between the chl$^{2-}$ anion and in situ formed **L¹** in the presence of zinc salts or cobalt salts did not yield trigonal bipyramidal cage products. This could be attributed to the mismatch between the size of the anionic template and the length of ligands. Instead, an elongated bis-formylpyridyl precursor ligand **B** (Fig. 3a) was designed. As anticipated, an enlarged trigonal bipyramidal structure, denoted as **5**, was successfully constructed. The ESI-MS spectrum of **5**, shown in Fig. 3c, d, displayed a series of +6 to +3 peaks corresponding to {[Co$^{II}$(chl)$_3$]⊂Co$^{II}_5$(**L³**)$_6$(ClO$_4$)$_{6-n}$}$^{n+}$ (where $n$ = 3, 4, 5, and 6). The structure was further confirmed by SCXRD analysis (Fig. 3b). Similar to **2-Cr**, $S$-**5** possesses an identical $D_3$-symmetric trigonal bipyramidal

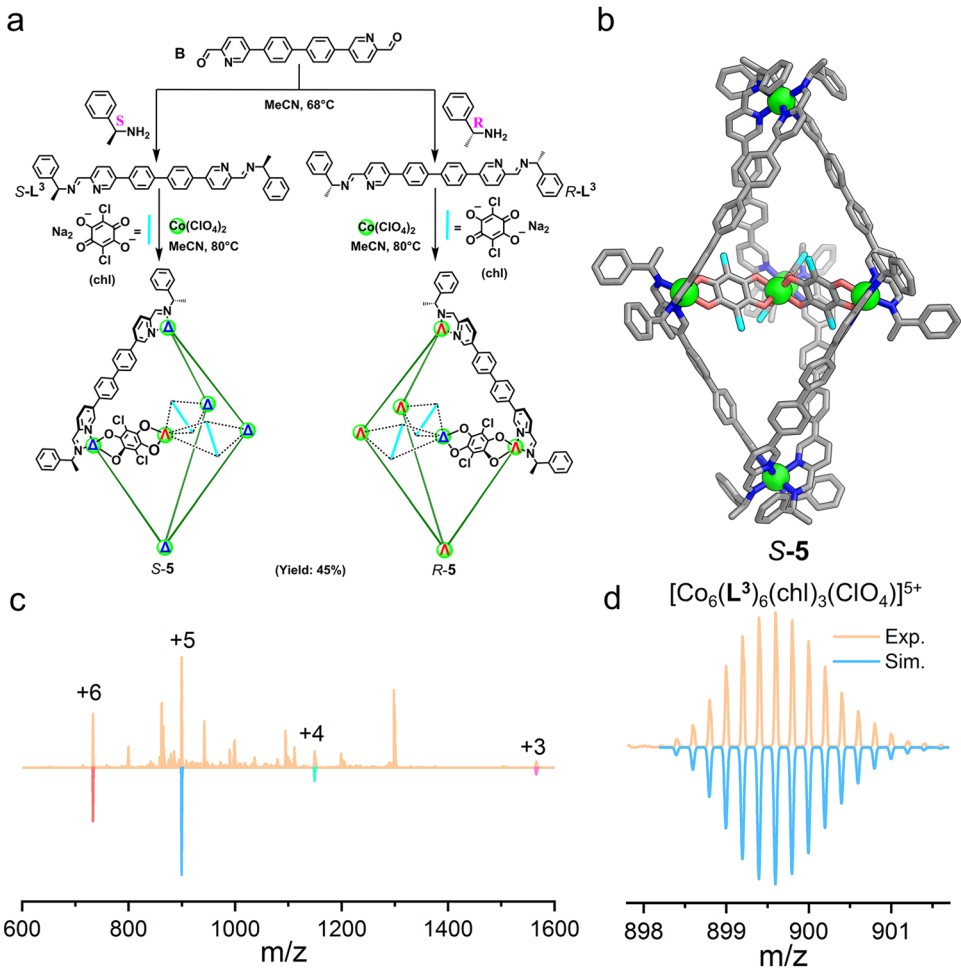

**Fig. 3 | Template synthesis with larger anions. a** The preparation of cage **5**. **b** X-ray crystal structures of cation parts of *S*-**5**, color codes: C, gray; N, blue; O, red; Cl, cyan; Co, green. Hydrogens were omitted for clarity. **c** ESI-MS spectrum of *S*-**5** with molecular peaks being labeled and compared with simulated peaks (downward ones). **d** Experimental and simulated isotopic patterns of 5+ charged molecular peak of **5**.

geometry but with a larger size. Three chloranilic anions self-organize around a cobalt ion to form a chiral-at-metal complex [Co$^{II}$(chl)$_3$]$^{4-}$, which is located in the center of the trigonal bipyramid. Two polar Co$^{II}$ ions are separated by 29.43 Å, and the edge length of the equatorial Co$^{II}_3$ triangle is 13.70 Å. The entire configuration of *S*-**5** is Λ-[Co$^{II}$(chl)$_3$]⊂Δ$_5$-Co$_5$(**L³**)$_6$, which is also defined by the chiral amine. CD studies further confirmed the diastereoselective synthesis of these cages (Supplementary Fig. 46). These experiments validate that $D_{2h}$-symmetric anions can effectively self-organize to template the assembly of trigonal bipyramids with suitable ligands, and the protocol is universal. It should be noted that the combination of ox²⁻ anions and in situ formed **L³** under the same condition was unable to result in trigonal bipyramidal cage products, highlighting the importance of the matching between the size of the template and the length of the ligand.

Four factors are crucial for the successful construction of these trigonal bipyramidal cages. The first consideration is the symmetry matching. The tetratopic $D_{2h}$-symmetric anions have the ability to in situ self-organize around an octahedral metal center to generate a $D_3$-symmetric chiral-at-metal complex, which is a hexatopic metalloligand to serve as a template. The ligation of the template into the framework of the host results in the final architecture expressing the symmetry of the template. The template and the host mutually stabilize and affect each other, since there is no isolated [Zn(ox)$_3$]$^{4-}$/[Zn(chl)$_3$]$^{4-}$ being documented and the chirality of the ligand was able to dictate the handedness of the template. The second aspect involves the optimal length ratio of the template and the ligand, allowing the chiral-at-metal template to be well-accommodated inside the host. The third factor is the Coulombic fit. The cationic cage and anionic template mutually attract each other. The ultimate factor lies in the stronger coordination ability of these anions than neutral ligands, enabling the preferential formation of $D_3$-symmetric chiral-at-metal templates.

The Δ-configuration of the [Zn(ox)$_3$]$^{4-}$/[Co(chl)$_3$]$^{4-}$ template is induced in the trigonal pyramidal cage by *R*-amine, which would be because of the geometrical matching between the Λ$_5$-M$_5$(*R*-L)$_6$ (M = Zn or Co) cage framework and the Δ-[Zn(ox)$_3$]$^{4-}$/[Co(chl)$_3$]$^{4-}$ ion, and vice versa. We try to propose a mismatched pair of Λ-Zn(ox)$_3$⊂Λ$_5$-Zn$_5$(*R*-L¹)$_6$ and optimize the structure by density functional theory (DFT) calculations. As shown in Supplementary Fig. 43, the mismatched pair of Λ-Zn(ox)$_3$⊂Λ$_5$-Zn$_5$(*R*-L¹)$_6$ complex accommodates the Λ-Zn(ox)$_3$ inside the Λ$_5$-Zn$_5$(*R*-L¹)$_6$ framework, requiring the distortion of oxalate ions. The structure optimization of the mismatched pair of Λ-Zn(ox)$_3$⊂Λ$_5$-Zn$_5$(*R*-L¹)$_6$ to reach energy minimum results in a matched pair of Δ-Zn(ox)$_3$⊂Λ$_5$-Zn$_5$(*R*-L¹)$_6$, consistent with the observation by SCXRD. The calculations indicate the matched pair of Δ-Zn(ox)$_3$⊂Λ$_5$-Zn$_5$(*R*-L¹)$_6$ is more energetically favorable compared to Λ-Zn(ox)$_3$⊂Λ$_5$-Zn$_5$(*R*-L¹)$_6$.

**Reversible photo-oxidation of anthracene-functionalized cages**
Molecular-based materials that exhibit CPL have attracted considerable interest owing to their wide-ranging potential applications[63]. Since **2-Zn** fluoresces upon radiation (Fig. 4e) and is enantiopure due to chiral inductions of chiral amines, we have to explore its CPL property. However, **2-Zn** showed negligible CPL properties (Supplementary

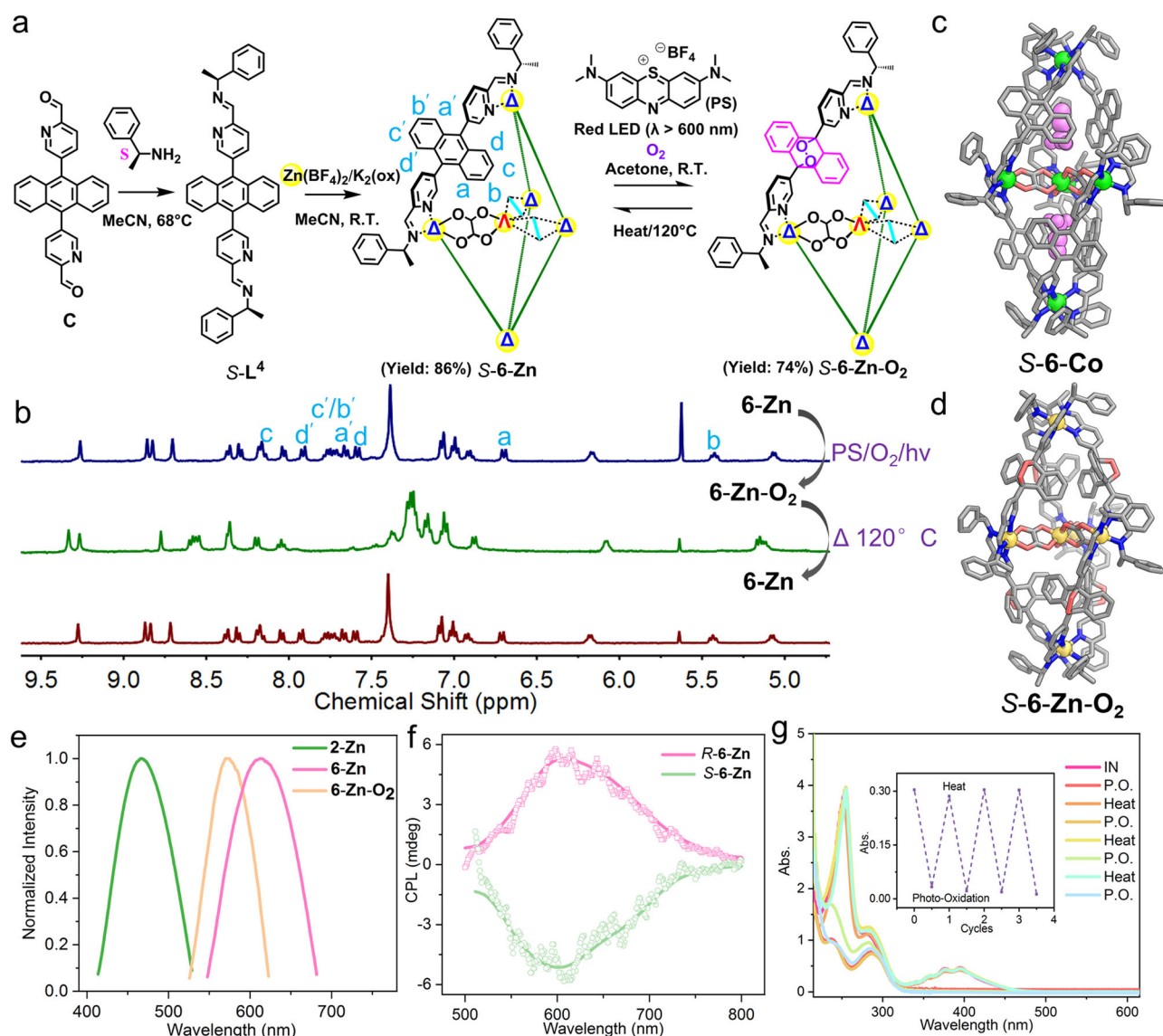

**Fig. 4 | Template synthesis of anthracene-functionalized cage and the reversible photo-oxygenation. a** The preparation of cage **6-Zn** and **6-Zn-O₂**. **b** Partial ¹H NMR spectra (acetone-d₆, 400 M, 298 K) of **6-Zn** (blue), after photo-oxidation to form **6-Zn-O₂** (green), and after heating of **6-Zn-O₂** at 120 °C overnight (red). X-ray crystal structures of cation parts of (**c**) S−**6-Co** and (**d**) S−**6-Zn-O₂**, color codes: C, gray; N, blue; O, red; Zn, yellow; Co, green; guest acetonitrile, violet. Hydrogens were omitted for clarity. **e** Luminescence spectra of **2-Zn**, **6-Zn** and **6-Zn-O₂** in the solid state. **f** CPL spectra of S−**6-Zn** and R−**6-Zn** in the solid state. **g** UV-vis absorption spectra showing reversible photo-oxidation of **6-Zn** with the inset illustrating the absorptions at 358 nm (samples were dissolved in acetonitrile; IN: the initial state of **6-Zn**; P.O.: after photo-oxidation, Heat: after heating).

Fig. 48). In order to improve the luminescent property and add additional functionalities to the cage, a bis-formylpyridyl precursor **C** (Fig. 4a) with an anthracene as a good chromophore was synthesized. Interestingly, without the addition of the oxalate templates, no tetrahedral cage product could be obtained by reacting Zn^II salts with subcomponent **C** and amines (Supplementary Fig. 32), presumably due to the congestion of neighboring anthracenes in forming an $M_4L^4_6$ tetrahedron. In contrast, trigonal bipyramidal cage **6-Zn** was successfully formed when potassium oxalates were added, which was verified by ¹H NMR (Fig. 4b) and ESI-MS (Supplementary Fig. 36). As shown in Fig. 4c, the crystal structure of S-**6-Co**, a congener of S-**6-Zn**, unambiguously confirmed the trigonal bipyramidal structure with the inclusion of a chiral-at-metal template. Due to the better enclosure of the cage provided by anthracenes, two acetonitrile molecules were found to be encapsulated in the twin cavities.

As shown in Fig. 4e, **6-Zn** in the solid state emitted red light with $\lambda_{max} = 613$ nm upon excitation by 456 nm, presenting a red shift of 147 nm compared to **2-Zn**. According to the crystal structure of **6-Co**, the anthracenes of the cage cannot freely rotate due to steric hindrances, which was also indicated by the non-equivalent proton signals of the two phenyl rings of the anthracenes in the ¹H NMR spectrum of **6-Zn** (Fig. 4b and Supplementary Fig. 33). This rotation restriction may reduce nonradiative relaxation upon excitation, making **6-Zn** a good fluorescent material. The ground-state frontier orbitals of **6-Zn** (Supplementary Fig. 44), as calculated by time-dependent density functional theory (TD-DFT), showed that the highest occupied molecular orbitals (HOMO) were primarily composed of orbitals from the anthracenes, while the lowest unoccupied orbitals were mainly distributed around one Zn^II-tris(pyridylimine) unit of the other end. The emission may originate from the ligand-ligand charge transfer (LLCT) related to the anthracenes and the Zn^II-tris(pyridylimine) unit, both of which are in a rigid chiral arrangement[64]. Consequently, as shown in Fig. 4f, S-**6-Zn** and R-**6-Zn** in the solid state demonstrated mirror CPL signals with the major peak around 610 nm, consistent with the

emission spectrum, reflecting their excited state chiral characteristics. The value of the anisotropy factor ($|g_{lum}|$) was $6.3 \times 10^{-4}$ for $S/R$-**6-Zn**, which was significantly smaller than chiral coordination cages constructed by rare earth metals[65–67] but comparable to those built with metals lacking f electrons[68,69]. The non-rotatable anthracenes fixed in a chiral configuration impose a rigid structure on **6-Zn**, thereby enhancing its chiroptical strength and inhibiting nonradiative relaxations upon excitation[70]. In contrast, the phenyl ring of the **L**$^1$ backbone in **2-Zn** could freely rotate, thus enabling more nonradiative relaxation upon excitation compared to the rigid structure of **6-Zn**. Consequently, **2-Zn** exhibits a lower quantum efficiency (5.3%, Supplementary Fig. 49) compared to **6-Zn** (15.4%). The lesser rigidity and lower quantum efficiency may potentially contribute to the absence of CPL properties in **2-Zn**.

The anthracenes, as functional moieties, are available for post-synthetic modifications by hetero-Diels-Alder reaction with singlet oxygen[71–74], which may be able to tune the fluorescent property[75]. The red light radiation ($\lambda > 600$ nm, 150 W) of **6-Zn** in an acetone solution, together with the photosensitizer 3,7-bis(dimethylamino)phenothiazinium tetrafluoroborate (**PS**, derivative of methylene blue) under an oxygen atmosphere, resulted in **6-Zn·O$_2$**. The ESI-MS spectrum of **6-Zn·O$_2$** clearly showed peaks for $\{[Zn^{II}(ox)_3] \subset Zn^{II}_5(\mathbf{L}^4)_6(O_2)_6(BF_4)_{6-n}\}^{n+}$ (where $n = 3$, 4, and 5), corroborating the incorporation of six $O_2$ molecules (Supplementary Fig. 42). The $^1$H NMR spectrum (Fig. 4b) of **6-Zn·O$_2$** demonstrated a neat profile belonging to one set of ligand signals, which is significantly different from that of **6-Zn**, hinting that all six anthracenes are quantitatively oxygenated to 9,10-anthracene endoperoxides (EPO). The crystal structure of **6-Zn·O$_2$** (Fig. 4d) undoubtedly confirmed the addition of six oxygens to all anthracenes of **6-Zn** to form bent 9,10-anthracene EPO derivatives. The EPO units directed outward from the cavities to form an all-out conformer. Anthracene-9,10-endoperoxide moieties were bent with a dihedral angle ($\varphi$) formed by the planes of the two aryl rings being about 120.9°. Compared with **6-Co**, the cavities of **6-Zn·O$_2$** were squeezed by the bent anthracene-9,10-endoperoxide moieties, of which one of the phenyl rings pointed toward the cavity. Consequently, no acetonitrile guest was accommodated inside **6-Zn·O$_2$** anymore, indicating that the photosensitized oxygenation reaction released the guest molecule trapped in the cavity. Although several photo-oxygenated cages have been reported[71–74], few of them have been characterized by X-ray crystallography; **6-Zn·O$_2$** represents a unique example of this type.

Upon heating the solid-state sample at 120 °C overnight, **6-Zn·O$_2$** was transformed back into **6-Zn**. These processes were also confirmed by $^1$H NMR (Fig. 4b) and UV-Vis spectra (Fig. 4g). As shown in Fig. 4b, the $^1$H NMR spectrum of **6-Zn·O$_2$** after heating is identical to that of fresh **6-Zn**, confirming the complete transformation back to **6-Zn**. In the absorption spectrum (Fig. 4g), **6-Zn** has fingerprint triple peaks in the range of 330–430 nm coming from the anthracenes, which disappeared when being oxygenated into **6-Zn·O$_2$**. After heating, these fingerprint triple peaks were restored to their initial values. The processes are fully reversible and could be repeated at least three times with good fatigue resistance. A portion of the released oxygen molecules were in the singlet state, which was confirmed by monitoring the absorption in the presence of $^1$O$_2$ trapping reagent 1,3-diphenylisobenzofuran (Supplementary Fig. 45)[72]. This indicates the cage may serve as a container for storage of $^1$O$_2$. $^1$O$_2$ is widely employed as an oxidant in synthetic organic chemistry, often produced through light activation with photosensitizers[76]. However, specific conditions may require light avoidance, such as when reacting with light-sensitive substrates. **6-Zn·O$_2$**, which contains $^1$O$_2$ and can undergo thermally-induced release, presents an alternative.

The emission of **6-Zn·O$_2$** is different from that of **6-Zn** (Fig. 4e), with $\lambda_{max} = 570$ nm upon excitation at 453 nm. CD studies confirmed that $S$-**6-Zn·O$_2$** and $R$-**6-Zn·O$_2$** are optically active (Supplementary Fig. 46). Nevertheless, these enantiopure $S$-**6-Zn·O$_2$** and $R$-**6-Zn·O$_2$**

failed to give effective CPL signals (Supplementary Fig. 48), presumably due to the low quantum efficiency (5.5%) (Supplementary Fig. 49)[77]. Interestingly, when they were transformed back to $S$-**6-Zn** and $R$-**6-Zn**, CPL signals of both were turned on (Supplementary Fig. 51). These experiments indicate that the on-off CPL signals of the system can be controlled through the reversible photo-oxygenation.

In conclusion, we discovered that $D_{2h}$ anions, such as oxalate anions and chloranilic anions, could self-organize around a metal ion, to form chiral-at-metal anionic complexes that can serve as templates for the construction of $D_3$-symmetric trigonal bipyramidal coordination cages. Chiral induction for the diastereoselective synthesis of these cages was accomplished using chiral amines, along with the stereoselective incorporation of the chiral-at-metal anionic template. This approach enabled the fabrication of various enantiopure trigonal bipyramidal cages including those adorned with functional anthracenes. Notably, the incorporation of anthracenes imparted CPL to the cages, which could be modulated through the reversible photo-oxygenation of the anthracenes. This study offers an approach to manufacturing metal-organic assemblies with precisely controlled geometry and functionalities by self-organized anions, thus giving unique insights into anion template synthesis. And the reversible photo-oxygenation affecting CPL properties of an assembly presented herein would also attract broad interest and provide potential applications, such as encryption[78,79].

## Methods

### Representative synthetic procedure

**cage 2.** Subcomponent **A** (10 mg, 34.7 μmol) and $R$−1-phenylethylamine (9 μL, 69.4 μmol) were dissolved in 2 mL acetonitrile. The mixture was stirred at 68 °C for 4 h. After cooling to room temperature, zinc tetrafluoroborate hydrate (12 mg, 34.7 μmol) and K$_2$C$_2$O$_4$ (3.2 mg, 17.3 μmol) which was dissolved in 30 μL water were added into the above solution. The mixture was stirred at room temperature overnight. The resulting solution was filtered into a thin tube, and anhydrous diethyl ether was layered onto the solution. The slow diffusion led to crystals of $R$-**2-Zn**. About 22.8 mg of crystalline cage $R$-**2-Zn** was obtained, yield: 92%. $S$-**2-Zn** was obtained by replacing $R$−1-phenylethylamine with $S$−1-phenylethylamine.

**cage 6-Zn·O$_2$.** Methylene blue hydrate (50 mg, 0.157 mmol) and AgBF$_4$ (340 mM, 0.98 mL, 66.4 μmol, 2.5 eq) were mixed in 25 mL of MeOH. After stirring for 1 h, the solvent was evaporated under vacuum to obtain a dark blue solid, which was further suspended in Et$_2$O (15.0 mL) and filtrated. The residual solid was washed with Et$_2$O (1 mL × 2) and extracted with CH$_2$Cl$_2$ (ca. 10 mL)[71]. About 170 mg of products (**PS**) was obtained, yield: 40%. **PS** (0.5 mg, 0.118 μmol, 0.1 eq.) was added into an acetone (600 μL) solution containing $R$-**6** (5.6 mg, 1.18 μmol) in a glass vessel equipped with a stir bar and capped with a rubber plug. The mixture was then filled with oxygen to keep it in an oxygen atmosphere while being irradiated with red lights using a LED lamp (SMD 5050 RGB, dominant wavelength range: 620–635 nm, $\lambda_{max} = 630$ nm) at room temperature for 16 hours. After that, the solvent was removed under vacuum. The crude product was dissolved in 0.5 mL acetone and excessive diethyl ether was added to precipitate. The precipitate was collected by centrifugation. The same processes were repeated by using dichloromethane, n-hexane, methanol instead of acetone, respectively, aiming at removing the photocatalyst. The final produce was dry under vacuum. About 4.3 mg of cage $R$-**6-Zn·O$_2$** was obtained, yield: 74%. $S$-**6-Zn·O$_2$** was obtained by replacing cage $R$-**6-Zn** with cage $S$-**6-Zn**.

The synthesis of **1**, **3**, **4**, **5**, and **6** is similar to that of **2**, and more details can be found in the Supplementary Information.

### Density functional theory (DFT) calculations

DFT calculations were conducted using the quantum chemistry program Gaussian 16[80]. A proposed mismatched pair of Λ-Zn(ox)$_3$⊂Λ$_5$-

$Zn_5(R\text{-}L^1)_6$ was based on the crystal structure of $R\text{-}2\text{-}Cr$ by changing the central $\Delta\text{-}Cr(ox)_3$ into $\Lambda\text{-}Zn(ox)_3$. Structural optimizations of $\Lambda\text{-}Zn(ox)_3 \subset \Lambda_5\text{-}Zn_5(R\text{-}L^1)_6$ were done at the B3LYP level, employing the 6−31 G(d) basis set for all elements. The initially guessed models of **3** and **4** were based on crystal structures of **1** and **2-Cr** according to the NMR and MS results. Structural optimizations of the models were first done on the PM3 level. The results were further optimized at the B3LYP level, employing the 6−31 G(d) basis set for C, H, N, O and Lanl2dz for Zn[81,82]. Structural optimizations of $S\text{-}6\text{-}Zn$ was based the crystal structure of it at the B3LYP level, employing the 6−31 G(d) basis set for all elements. Time-dependent DFT calculations were conducted using the same function and basis set. Forty singlet states (nstates = 40, singlet) were chosen in the calculations.

## Crystallographic data and structure refinement

The crystals were fragile and easily lost solvent molecules. Thus low temperature and rapid handing of the samples were needed. Diffraction data of $R\text{-}1$, $R\text{-}2\text{-}Cr$ and $S\text{-}6\text{-}Co$ were collected on Bruker SMART APEX II X-Ray diffractometer at 150 K using Mo-K$\alpha$ ($\lambda = 0.71073$ Å) X-ray sources. Data were processed with the INTEGRATE program of the APEX2 software for reduction and cell refinement. Multi-scan absorption corrections were applied by using the SCALE program for area detector. Diffraction data of $S\text{-}5$ was collected on a Rigaku XtaLAB Synergy (DW system, HyPix) X-Ray diffractometer at 100 K using micro-focus X-ray sources (Cu K$\alpha$, $\lambda = 1.54184$ Å). Data were processed with *CrysAlisPro* software suite. Empirical absorption correction was done by using spherical harmonics, implemented in SCALE3 ABSPACK scaling algorithm. Data of $S\text{-}6\text{-}Zn\text{-}O_{12}$ were recorded on Bruker D8 VENTURE diffractometer with an Incoatec I$\mu$S 3.0 Cu EF microfocus source (55 W, Cu K$\alpha$, $\lambda = 1.54178$ Å) equipped with a PHOTON III C28 detector at 100 K. The raw frame data were processed using SAINT and SADABS to yield the reflection data file.

Using Olex2[83], the structures were solved with the ShelXT structure solution program using Intrinsic Phasing and refined with the ShelXL refinement package using Least Squares minimization[84]. In general, non-hydrogen atoms with occupancies greater than 0.5 were refined anisotropically. Carbon-bound hydrogen atoms were included in idealized positions and refined using a riding model. Crystallographic data have been deposited with the CCDC: 2307653 ($R\text{-}1$), 2307651 ($R\text{-}2\text{-}Cr$), 2307652 ($S\text{-}5$), 2112669 ($S\text{-}6\text{-}Co$) and 2307654 ($S\text{-}6\text{-}Zn\text{-}O_2$). More details could be found in Supplementary Information.

## Data availability

The authors declare that all data supporting the findings of this study are included within the article and its Supplementary Information, and are also available from the authors upon request. Correspondence and requests for materials should be addressed to Y.Y. or D.Z. Crystallographic data for the structures reported in this paper have been deposited at the Cambridge Crystallographic Data Centre, under the deposition numbers 2307653 ($R\text{-}1$), 2307651 ($R\text{-}2\text{-}Cr$), 2307652 ($S\text{-}5$), 2112669 ($S\text{-}6\text{-}Co$) and 2307654 ($S\text{-}6\text{-}Zn\text{-}O_2$). Copies of these data can be obtained free of charge via www.ccdc.cam.ac.uk/data_request/cif. Atomic coordinates of the optimized computational models have been provided in the Supplementary Data.

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

## Acknowledgements

Y.Y. acknowledges supports of the National Natural Science Foundation of China (No. 22071083) and Natural Science Foundation of Jiangsu Province (Grant BK20211608). D.Z. is thankful to the National Natural Science Foundation of China (No.22201075).

## Author contributions

Y.Y. and D.Z. conceived the study, analyzed the results and wrote the manuscript. S.G., W.Z., and F.Y. contributed equally in performing the experiments, collecting the X-ray data. Y.Y. refined the structures. Y.D. and J.Z. performed parts of the experiments. All authors discussed the results and edited the manuscript.

## Competing interests

The authors declare no competing interests.
