## [Peer Review File · Nature Communications]

Enantiopure trigonal bipyramidal coordination cages templated by in situ self-organized D_{2h}-symmetric anionsReviewers' Comments:

Reviewer #1:

Remarks to the Author:

This paper reports the self-assembly of trigonal bipyramidal shaped chiral metal-organic cages induced by trioxalate-functionalized mononuclear metal center. Well-known tetrahedral Zn(II) metal-organic cages by sub-component approach were converted into novel trigonal bipyramidal metal-organic cages by addition of oxalates with transition metal ion. The structures of trigonal bipyramidal metal-organic cages were characterized by ESI-MS, ¹H NMR, and single-crystal X-ray analyses (some of them). The main significance of this report is the formation of novel trigonal pyramidal cage by anionic ligand such as oxalate. This approach has potential impact in the field of metal-organic cages. As to other results (CPL property, reaction of singlet oxygen with ligand possessing anthanthrene), related reports have been published before, so these results are optional to increase the data. Following are the comments that would deepen the discussion and improve the manuscript.

1. The authors propose three reasons why oxalate induced the formation of the trigonal pyramidal cage in the manuscript. I wonder if the coordination strength of oxalate is stronger than that of pyridylimine and 2,2-bpy. If oxalate is stronger ligand, the addition of oxalate causes to make coordination bonds between Zn and oxalate, breaking the coordination bonds between pyridylimine and Zn ion, which is the driving force of the formation of the trigonal pyramidal cage. To clarify this possibility, simple questions are

(1) What happened when oxalate is added in the tetrahedral cage without addition of additional Zn ions? If nothing happens, the coordination ability of oxalate is weaker than that of pyridylimine, and the possibility of stronger coordination ability of oxalate to induce the trigonal pyramidal cage is excluded.

(2) What happened when oxalate is titrated in a solution of the tetrahedral cage until excess amount of oxalate is added?

(3) Is it possible to protonate the oxalate ligands in the trigonal pyramidal cage? If this is possible, the authors can demonstrate the interconversion between the tetrahedral and trigonal pyramidal cages by acid-base control.

2. Δ-configuration is induced in the trigonal pyramidal cage by s-amine, which would be because of the geometrical matching between the two types of chiral centers (ΛZn- pyridylimines and Δ-configuration in Zn-oxalates). The structural difference between matched and mismatched pair of the trigonal pyramidal cage by theoretical calculations can support this idea and may clarify the reason why the combination of ΛZn- pyridylimines and Δ-configuration in Zn-oxalates is the matched pair.

3. Why was the CPL property observed in the trigonal bipyramidal cage 6-Zn by introduction of anthracene unit in the ligand, whereas the trigonal bipyramidal cage without anthracene (2-Zn) did not show CLP property?

4. The anisotropy factor of 6-Zn should be compared with those for other molecules and the authors should discuss why 6-Zn showed such high or low anisotropy factor based on its molecular structure.

5. As to the interconversion between 6-Zn and 6-Zn-O₂, the authors state that "the cage may serve as a container for storage of 1O₂." in the manuscript. In what areas can this technology be used, what are the requirements for the practical use, and does the present system (6-Zn and 6-Zn-O₂) satisfy these requirements?

Reviewer #2:

Remarks to the Author:

This review only concerns crystallographic aspects of the work.

Some revisions to the crystallographic refinements of the structures and contents of the CIFs are requested before publication can be recommended. For the most part the determination of these challenging structures has been carried out diligently, however, some clarifications and improvements are required. The referee particularly appreciates the author's use of the `_refinement_special_details` section of the CIF to discuss decisions made during the refinement.

The following points and requests apply to all structures in the manuscript.

Where SQUEEZE or a mask has been applied to handle diffuse solvent and anions an attempt should be made to estimate the residues omitted and include them in both the sum formula (SFAC/UNIT line in the res file) and chemical formula moiety (Squeezed or masked residues to be appended in square brackets). This estimation should be made on the basis of charge balance, estimated electron count and void volumes (the Mask function in Olex2 is very helpful in this job). It can reasonably be assumed that any Squeezed moieties other than counterions indicated by ESI are acetonitrile solvent residues.

All structures should make use of the rigid bond restraint RIGU in SHELXL for all anisotropic non-metal atoms. Where the data-parameter ratio is necessarily low because of weak diffraction it is entirely appropriate and helpful to apply this restraint widely (this restraint should be used in place of the outdated restraint DELU which is used in S-6-Co). Use of this restraint may allow the severity of the SIMU restraint to be reduced: in some of the structures the SIMU restraint has been used with very small esd values which might be permitted to be relaxed).

The choice to use a rigid hexagon constraint (AFIX 66) for many phenyl and pyridine rings in the structure should be discussed and probably revisited. The rigid hexagon restraint is chemically inappropriate for these groups (especially the pyridine rings) and the preferred refinement method should be to use restraints (DFIX or SADI) to allow the structure to refine to a chemically sensible model. The presence of multiple chemically identical moieties in all the structures means that wider use of the similar geometry restraint (SADI) can be applied to chemically similar 1,2 and 1,3 distances. Better still, the SAME command should be applied to chemically identical moieties to automatically apply such geometric similarity restraints (SADI) to all 1,2 and 1,3 bond distances. If this more appropriate restraint approach is not adopted the reasons should be discussed in the refinement special details.

In the CIFs the field `_chemical_absolute_configuration` should be correctly filled in (check the CIF dictionary for permitted entries in this field. Where the Flack/Hooft parameters are within 3 esd of zero then 'AD' would be an appropriate entry to conform the absolute structures were conformed by anomalous dispersion. Otherwise the description 'SYN' would be appropriate and discussion should be included in the refinement_special_details about what alternative analysis supports the presented configuration. Where the absolute structure as determined by crystallography in the manuscript text the Flack or Hooft parameters and their esd should be quoted. Where the Flack or Hooft parameter is significantly non-zero the possibility of inversion twinning should be discussed – can the authors be certain that it isn't possible for these structures?

Points that relate to specific structures:

The geometries of the perchlorate anion (S-6-Co) and the triflimide anions (S-6-Zn-O12) should be restrained (DFIX, SADI or SAME) to have more chemically appropriate geometries.

In structure S-6-Zn-O12 the thermal parameters of phenyl rings C9, C140, C188 C285 are far larger than can be accounted for solely on the basis of increased thermal motion (several chemically similar phenyl moieties in the structure have much more normal values). An attempt should be made to model disorder and the outcome of this attempt discussed in the refinement special details. I would suggest that these phenyl rings refine more appropriately at half occupancy – what might this observation mean for any conclusions drawn about the configuration of the adjacent chiral centre?

Reviewer #3:

Remarks to the Author:

Accept with minor revision. The authors reported a rare example of templating effects introduced by D_{2h}-symmetric anions with different sizes on the formations of enantiopure trigonal bipyramidal coordination cages assembled with various ligands respectively. The experiments are sound and the conducted control experiments supports the anionic templating effects, with MS, NMR, DFT and X-ray data. This method was further used to synthesize cages with interesting properties such as CPL and even photo-oxygenated properties, showing the potential applications of such type supramolecular assemblies.

Some small points to address:-

1. The CD spectra can be added to the section of Photophysical properties for convenient reading
2. For the solubility issue of ¹³C nmr, it is suggested that the authors can try other d-solvents to obtain better carbon nmr, e.g. for FigS 41 and ¹³C nmr for synthesis section 2.2.

3. It is suggested that the goodness of fit can be added to the lifetime measurement of 6-Zn. The author should also state that whether the curve is monoexponential decay or not, to prove there is single species in 6-Zn.
4. MS data for intermediates are lacking.
5. Fig. 4g, what is the condition for these UV measurements, solid state?
6. The sequency of supplementary figure in manuscript should rearrange in a better order. E.g. line 247, FigS.50, line 288-291, FigS43-48.
7. Fig 1a: it is better to add relative equivalent of the materials. It is difficult to realize whether excess of Zn²⁺ was added in the reaction mixture or just a suitable amount was added instead.
(the figures like this one at times are over cluttered – maybe the reaction schemes should be separated from the crystal structures – (the crystals can also have the hydrogens omitted for clarity) and also have some of the symmetry elements indicated.
8. [ox] is using in the paragraph but [C2O4] is using in the scheme. They should be consistent in the whole manuscript.
9. Line 118: it should be (b) R-1.
10. References list: for example in reference 1, it should be Bols, P. S., Anderson, H. L. etc

Response to Referees

Reviewer #1 (Remarks to the Author):

This paper reports the self-assembly of trigonal bipyramidal shaped chiral metal-organic cages induced by trioxalate-functionalized mononuclear metal center. Well-known tetrahedral Zn(II) metal-organic cages by sub-component approach were converted into novel trigonal bipyramidal metal-organic cages by addition of oxalates with transition metal ion. The structures of trigonal bipyramidal metal-organic cages were characterized by ESI-MS, ¹H NMR, and single-crystal X-ray analyses (some of them). The main significance of this report is the formation of novel trigonal pyramidal cage by anionic ligand such as oxalate. This approach has potential impact in the field of metal-organic cages. As to other results (CPL property, reaction of singlet oxygen with ligand possessing anthanthrene), related reports have been published before, so these results are optional to increase the data. Following are the comments that would deepen the discussion and improve the manuscript.

Response: We thank the positive comments and kind suggestions from the reviewer.

1. The authors propose three reasons why oxalate induced the formation of the trigonal pyramidal cage in the manuscript. I wonder if the coordination strength of oxalate is stronger than that of pyridylimine and 2,2-bpy. If oxalate is stronger ligand, the addition of oxalate causes to make coordination bonds between Zn and oxalate, breaking the coordination bonds between pyridylimine and Zn ion, which is the driving force of the formation of the trigonal pyramidal cage. To clarify this possibility, simple questions are

(1) What happened when oxalate is added in the tetrahedral cage without addition of additional Zn ions? If nothing happens, the coordination ability of oxalate is weaker than that of pyridylimine, and the possibility of stronger coordination ability of oxalate to induce the trigonal pyramidal cage is excluded.

(2) What happened when oxalate is titrated in a solution of the tetrahedral cage until excess amount of oxalate is added?

(3) Is it possible to protonate the oxalate ligands in the trigonal pyramidal cage? If this is possible, the authors can demonstrate the interconversion between the tetrahedral and trigonal pyramidal cages by acid-base control.

Response: We thank the reviewer for the kind suggestions and the reviewer is very insightful. We have performed experiments of titrating potassium oxalate into the tetrahedral cage *R-1* without the addition of additional Zn ions. As shown in **Fig. R1** (Supplementary **Fig. 19**), without the additional Zn ions, the sole oxalate ions lead to the decomposition of *R-1* cage resulting the only ligand **L¹**. In

comparison, as shown in **Fig. R2** (Supplementary **Fig. 18**), with 2 equivalent $\text{Zn}(\text{BF}_4)_2$ and 3 equivalent potassium oxalate, the *R-1* gradually transformed into *R-2-Zn*. According to these experiments, the oxalate ion is a stronger ligand than the pyridylimine. The breaking of the coordination bonds between pyridylimine and the Zn ion by oxalate ions is the driving force of the formation of the trigonal pyramidal cage. We have added *“Additionally, we attempted to titrate potassium oxalates into solution of 1 until excess without additional Zn salts, resulting in the decomposition of 1 into free ligands (Supplementary Fig. 19). These observations suggest that the oxalate ion is a stronger ligand than the pyridylimine. The introduction of oxalate ions promoted the formation of coordination bonds between Zn^{II} ions and oxalate ions, disrupting the coordination bonds between pyridylimine and Zn^{II} ions, potentially serving as a driving force for the transformation.”* in the manuscript. *“The ultimate factor lies in the stronger coordination ability of these anions than neutral ligands, enabling the preferential formation of D_3 -symmetric chiral-at-metal templates.”* has also been added into the discussion of factors that are crucial for the successful construction of these trigonal bipyramidal cages.

Fig. R1 (Supplementary **Fig. 19**). ^1H NMR spectra (CD_3CN , 298K) of additions of different mole amounts of potassium oxalate into the solution of *R-1* with *R-1*, *R-2-Zn* and L^1 as references. The addition of potassium oxalate without additional Zn^{2+} ions resulted in decomposition of tetrahedral

cage **1**, indicating stronger coordination ability of oxalate ion than ligands with pyridylimines.

Fig. R2 (Supplementary **Fig. 18**). ¹H NMR spectra (CD₃CN, 298K) of time-dependent reaction mixture containing *R-1* and 2 eq. Zn(BF₄)₂ and 3 eq. potassium oxalate with *R-1*, *R-2-Zn* as references.

We have also tried to add small amount of trifluoroacetic acid (TFA) into the acetonitrile solution of *R-2-Zn*, attempting to protonate the oxalate ligands. The result is shown in **Fig. R3**. However, the imine bonds are sensitive to acid, which will hydrolyze to result in the decomposition of the cage. Therefore, it is unable to achieve the interconversion between the tetrahedral and trigonal pyramidal cages by acid-base control.

Fig. R3. ^1H NMR spectra (CD_3CN , 298K) of *R-2-Zn* and after the addition of small amount (3 eq. of the cage) of trifluoroacetic acid (TFA). The results indicate the decomposition of *R-2-Zn* in the presence of acid.

2. Δ -configuration is induced in the trigonal pyramidal cage by *s*-amine, which would be because of the geometrical matching between the two types of chiral centers (Δ -Zn- pyridylimines and Δ -configuration in Zn-oxalates). The structural difference between matched and mismatched pair of the trigonal pyramidal cage by theoretical calculations can support this idea and may clarify the reason why the combination of Δ -Zn-pyridylimines and Δ -configuration in Zn-oxalates is the matched pair.

Response: We thank the reviewer for the kind suggestions. We have added a discussion “The Δ -configuration of the $[\text{Zn}(\text{ox})_3]^{4+}/[\text{Co}(\text{chl})_3]^{4+}$ template is induced in the trigonal pyramidal cage by *R*-amine, which would be because of the geometrical matching between the $\Delta_5\text{-M}_5(\text{R-L})_6$ ($M = \text{Zn}$ or Co) cage framework and the Δ - $[\text{Zn}(\text{ox})_3]^{4+}/[\text{Co}(\text{chl})_3]^{4+}$ ion, and vice versa. We try to propose a mismatched pair of Δ - $\text{Zn}(\text{ox})_3 \subset \Delta_5\text{-Zn}_5(\text{R-L}^1)_6$ and optimize the structure by theoretical calculations. As shown in Supplementary **Fig. 42**, the mismatched pair of Δ - $\text{Zn}(\text{ox})_3 \subset \Delta_5\text{-Zn}_5(\text{R-L}^1)_6$ complex accommodates the Δ - $\text{Zn}(\text{ox})_3$ inside the $\Delta_5\text{-Zn}_5(\text{R-L}^1)_6$ framework, requiring the distortion of oxalate ions. The structure optimization of the mismatched pair of Δ - $\text{Zn}(\text{ox})_3 \subset \Delta_5\text{-Zn}_5(\text{R-L}^1)_6$ to reach energy”

minimum results in a matched pair of Δ -Zn(ox)₃⊂Λ₅-Zn₅(R-L¹)₆, consistent with the observation by single crystal X-Ray diffraction. The calculations indicate the matched pair of Δ -Zn(ox)₃⊂Λ₅-Zn₅(R-L¹)₆ is more energetically favorable compared to Δ -Zn(ox)₃⊂Λ₅-Zn₅(R-L¹)₆.” into the manuscript.

Fig. R4 (Supplementary **Fig. 42**). The structure optimization of a mismatched pair of Λ -Zn(ox)₃⊂Λ₅-Zn₅(R-L¹)₆ using Gaussian 16 with B3LYP function and 6-31G(d) basis set. The initial state is a Λ -Zn(ox)₃⊂Λ₅-Zn₅(R-L¹)₆, while energy minimum state after structural optimization is a Δ -Zn(ox)₃⊂Λ₅-Zn₅(R-L¹)₆.

3. Why was the CPL property observed in the trigonal bipyramidal cage 6-Zn by introduction of anthracene unit in the ligand, whereas the trigonal bipyramidal cage without anthracene (2-Zn) did not show CLP property?

Response: We thank the reviewer for the kind suggestions. Numerous factors influence the CPL property of a compound, making the precise determination challenging. A previous report (J.-X. Tang et. al. *Adv. Optical Mater.* **2021**, 9, 2100017, Ref.70) indicated the structure rigidity help to get better CPL property. And quantum efficiency also stands out as a crucial factor. We have added a short discussion *“The non-rotatable anthracenes fixed in a chiral configuration impose a rigid structure on 6-Zn, thereby enhancing its chiroptical strength and inhibiting nonradiative relaxations upon excitation [Ref. 70]. In contrast, the phenyl ring of the L¹ backbone in 2-Zn could freely rotate.*

thus enabling more nonradiative relaxation upon excitation compared to the rigid structure of 6-Zn. Consequently, 2-Zn exhibits a lower quantum efficiency (5.3%, Supplementary Fig. 48) compared to 6-Zn (15.4%). The lesser rigidity and lower quantum efficiency may potentially contribute to the absence of CPL properties in 2-Zn.” into the manuscript.

4. The anisotropy factor of 6-Zn should be compared with those for other molecules and the authors should discuss why 6-Zn showed such high or low anisotropy factor based on its molecular structure.

Response:

Response: We thank the reviewer for the kind suggestions. Only a limited numbers of coordination cages have been reported to have CPL property in solid state. The g_{lum} factor of 6-Zn is significant smaller than chiral coordination cages constructed by rare earth metals (Ref. 65-67), while comparable to those constructed by Pd^{2+} , Zn^{2+} and Cu^+ ions which were also lack of f electrons (Ref. 68, 69). We have added “The value of the anisotropy factor ($|g_{lum}|$) was 6.3×10^{-4} for S/R-6-Zn, which was significantly smaller than chiral coordination cages constructed by rare earth metals [Ref. 65-67] but comparable to those built with metals lacking f electrons [Ref. 68, 69].” into the manuscript

5. As to the interconversion between 6-Zn and 6-Zn-O₂, the authors state that “the cage may serve as a container for storage of ¹O₂.” in the manuscript. In what areas can this technology be used, what are the requirements for the practical use, and does the present system (6-Zn and 6-Zn-O₂) satisfy these requirements?

Response: We thank the reviewer for the kind suggestions. ¹O₂ is widely employed as an oxidant in synthetic organic chemistry, often produced through light activation with photosensitizers. However, specific conditions may require light avoidance, such as when reacting with light-sensitive substrates. 6-Zn-O₂, which contains ¹O₂ and can undergo thermally-induced release, presents an alternative. These applications necessitate the release of ¹O₂ upon simulation and easy removal of the cage after reacting. The present system basically satisfies the requirement. We have added “¹O₂ is widely employed as an oxidant in synthetic organic chemistry, often produced through light activation with photosensitizers. However, specific conditions may require light avoidance, such as when reacting with light-sensitive substrates. 6-Zn-O₂, which contains ¹O₂ and can undergo thermally-induced release, presents an alternative.” into the manuscript.

Reviewer #2 (Remarks to the Author):

This review only concerns crystallographic aspects of the work.

Some revisions to the crystallographic refinements of the structures and contents of the CIFs are requested before publication can be recommended. For the most part the determination of these challenging structures has been carried out diligently, however, some clarifications and improvements are required. The referee particularly appreciates the author's use of the `_refinement_special_details` section of the CIF to discuss decisions made during the refinement.

Response: We thank the positive comments and kind suggestions from the reviewer. We have added discussions of decisions for refinement in the '`_refinement_special_details`' sections of all the CIF.

The following points and requests apply to all structures in the manuscript.

1. Where SQUEEZE or a mask has been applied to handle diffuse solvent and anions an attempt should be made to estimate the residues omitted and include them in both the sum formula (SFAC/UNIT line in the res file) and chemical formula moiety (Squeezed or masked residues to be appended in square brackets). This estimation should be made on the basis of charge balance, estimated electron count and void volumes (the Mask function in Olex2 is very helpful in this job). It can reasonably be assumed that any Squeezed moieties other than counterions indicated by ESI are acetonitrile solvent residues.

Response: We thank the reviewer for the kind suggestions. The unidentified counterions are estimated by charge balance, which is also supported by ESI-MS studies. The amounts of acetonitrile solvent residues have been estimated by the electron count and void volumes suggested by SQUEEZE, by assuming that any Squeezed moieties other than counterions indicated by ESI-MS are acetonitrile solvent residues. The amounts of acetonitrile solvent and the unidentified counterions have been included in both the sum formula (SFAC/UNIT line in the res file) and chemical formula moiety (Squeezed or masked residues to be appended in square brackets). The details have been added to '`_refinement_special_details`' in the CIF. We have also revised the "Crystallographic data and structure refinement" section of the Supplementary Information.

2. All structures should make use of the rigid bond restraint RIGU in SHELXL for all anisotropic non-metal atoms. Where the data-parameter ratio is necessarily low because of weak diffraction it is entirely appropriate and helpful to apply this restraint widely (this restraint should be used in place of the out-dated restraint DELU which is used in S-6-Co). Use of this restraint may allow the severity of the SIMU restraint to be reduced: in some of the structures the SIMU restraint has been

used with very small esd values which might be permitted to be relaxed).

Response: We thank the reviewer for the kind suggestions. We have tried to widely use RIGU restraints. The DELU which was used in *S-6-Co* has been removed. Most of The SIMU restraints with very small esd values were allow to relax by applications of RIGU restrains. The R1 value of *R-1* has been also reduced from 9.43% to 9.24%, and that of *R-2-Cr* has been reduced from 8.61% to 8.46%.

3. The choice to use a rigid hexagon constraint (AFIX 66) for many phenyl and pyridine rings in the structure should be discussed and probably revisited. The rigid hexagon restraint is chemically inappropriate for these groups (especially the pyridine rings) and the preferred refinement method should be to use restraints (DFIX or SADI) to allow the structure to refine to a chemically sensible model. The presence of multiple chemically identical moieties in all the structures means that wider use of the similar geometry restraint (SADI) can be applied to chemically similar 1,2 and 1,3 distances. Better still, the SAME command should be applied to chemically identical moieties to automatically apply such geometric similarity restraints (SADI) to all 1,2 and 1,3 bond distances. If this more appropriate restraint approach is not adopted the reasons should be discussed in the refinement special details.

Response: We thank the reviewer for the kind suggestions. We have removed all the AFIX 66 constraints for the pyridine rings and anthracenes. SADI have been used for these rings to get reasonable geometries.

4. In the CIFs the field `_chemical_absolute_configuration` should be correctly filled in (check the CIF dictionary for permitted entries in this field. Where the Flack/Hooft parameters are within 3 esd of zero then 'AD' would be an appropriate entry to conform the absolute structures were conformed by anomalous dispersion. Otherwise the description 'SYN' would be appropriate and discussion should be included in the refinement_special_details about what alternative analysis supports the presented configuration. Where the absolute structure as determined by crystallography in the manuscript text the Flack or Hooft parameters and their esd should be quoted. Where the Flack or Hooft parameter is significantly non-zero the possibility of inversion twinning should be discussed – can the authors be certain that it isn't possible for these structures?

Response: We thank the reviewer for the kind suggestions. We have revised this '`_chemical_absolute_configuration`' value by adding 'AD' for *S-5* and *S-6-Co*, and 'SYN' for *R-1*, *R-2-Cr* and *S-6-Zn-O₁₂*. Because the Flack factors for *R-1*, *R-2-Cr* and *S-6-Zn-O₁₂* are relatively large due to the limit resolution and high levels of thermal motion of chiral moieties of ligands. We

have removed the quotations of Flack in the text, since some of them are relatively large. However, the configuration of them are supported by the solution CD studies of these samples, which all showed active signals. Furthermore, chiral amines used for synthesis of these cages are enantiopure and with known exact handedness, which are also consistent with absolute structures as determined by crystallography. Thus, we believe there is no inversion twinning. We have added "*The configuration of the crystal was supported by the solution CD studies of bulk samples, which showed active signals. Furthermore, chiral amines used for synthesis of the cages are enantiopure and with known exact handedness, which are also consistent with absolute structures as determined by crystallography. Thus, we believe there is no inversion twinning.*" into the '_refinement_special_details' sections of all the CIF.

Points that relate to specific structures:

5. The geometries of the perchlorate anion (S-6-Co) and the triflimide anions (S-6-Zn-O12) should be restrained (DFIX, SADI or SAME) to have more chemically appropriate geometries.

Response: We thank the reviewer for the kind suggestions. We have revised accordingly by applying DFIX and SADI to refine the counterions into chemically appropriate geometries.

6. In structure S-6-Zn-O12 the thermal parameters of phenyl rings C9, C140, C188 C285 are far larger than can be accounted for solely on the basis of increased thermal motion (several chemically similar phenyl moieties in the structure have much more normal values). An attempt should be made to model disorder and the outcome of this attempt discussed in the refinement special details. I would suggest that these phenyl rings refine more appropriately at half occupancy – what might this observation mean for any conclusions drawn about the configuration of the adjacent chiral centre?

Response: We appreciate the reviewer's valuable suggestions. We concur that the thermal parameters of phenyl rings C140, C188, C285, and C236 exceed those of similar phenyl groups in the structure, which exhibit more typical values. We attempted to consider these phenyl rings as disordered at half occupancy but were unsuccessful in identifying suitable alternate half occupancies, or any other occupation values. Any attempts to model an alternative position for these rings result in configurations with significantly large thermal parameters, despite the application of strict restraints. We found that the phenyl groups with normal thermal parameters all form obviously π - π interactions with adjacent ligands, while the ones with large thermal parameters are on the tail of the ligands and able to dynamically rotate without forming any intra-molecular or inter-molecular interactions with neighbours. Thus, it is reasonable for these phenyl groups having larger thermal parameters than chemically similar phenyl moieties. By applying combination of RIGU and SIMU, the revised data

of **S-6-Zn-O₁₂** show slightly decreased thermal parameters of phenyl rings C140, C188, C285, and C236 as compared with the previous one. We have added “The thermal parameters of phenyl rings C140, C188, C285, and C236 exceed those of similar phenyl groups in the structure, which exhibit more typical values. We attempted to consider these phenyl rings as disordered at half occupancy, or any other occupation values, but were unsuccessful in identifying suitable alternate half occupancies. Any attempts to model an alternative position for these rings result in configurations with significantly large thermal parameters, despite the application of strict restraints. We found that the phenyl groups with normal thermal parameters all form obviously π - π interactions with adjacent ligands, while the ones with large thermal parameters are on the tail of the ligands and able to dynamically rotate without forming any intra-molecular or inter-molecular interactions with neighbors. Thus, it is reasonable for these phenyl groups to have larger thermal parameters than chemically similar phenyl moieties.” in the ‘_refinement_special_details’ sections of the CIF. The high levels of thermal motion of chiral moieties of ligands (including these tail phenyl rings, which bond to the chiral C centers) may relate to the large Flack factor of this data. However, the enantiopurity was supported by CD studies of the crystal samples. The chiral amine used for synthesis of this cage is enantiopure and with known exact handedness, which is consistent with the absolute structure as determined by crystallography.

Reviewer #3 (Remarks to the Author):

Accept with minor revision. The authors reported a rare example of templating effects introduced by D_{2h}-symmetric anions with different sizes on the formations of enantiopure trigonal bipyramidal coordination cages assembled with various ligands respectively. The experiments are sound and the conducted control experiments supports the anionic templating effects, with MS, NMR, DFT and X-ray data. This method was further used to synthesize cages with interesting properties such as CPL and even photo-oxygenated properties, showing the potential applications of such type supramolecular assemblies.

Response: We thank the positive comments and kind suggestions from the reviewer.

Some small points to address:

1. The CD spectra can be added to the section of Photophysical properties for convenient reading

Response: We thank the review’s kind suggestion. We have revised accordingly. A new

Supplementary **Fig. 45** with all the CD spectra has been added into the section of Photophysical properties.

2. For the solubility issue of ^{13}C nmr, it is suggested that the authors can try other d-solvents to obtain better carbon nmr, e.g. for FigS 41 and ^{13}C nmr for synthesis section 2.2.

Response: We thank the review's kind suggestion. The suggestion worked well for ligand **B**. We have tried CD_2Cl_2 as solvent for ligand **B** in section 2.2, which resulted in a better ^{13}C NMR spectrum for it. We have provided a new ^{13}C NMR spectrum for ligand **B** in the Supplementary Information as Supplementary **Fig. 4**. However, we have tried several other solvents, including CD_3CN , CD_3OD and CD_3NO_2 , for dissolving crystalline **6-Zn-O₂** to obtain better ^{13}C NMR spectrum, which all failed.

3. It is suggested that the goodness of fit can be added to the lifetime measurement of 6-Zn. The author should also state that whether the curve is monoexponential decay or not, to prove there is single species in 6-Zn.

Response: We thank the review's kind suggestion. We have added the goodness of fitting for the lifetime measurement for **6-Zn** (Supplementary **Fig. 49**). We have added "*It is monoexponential decay, which indicates single species in 6-Zn.*" in the caption of Supplementary **Fig. 49**.

4. MS data for intermediates are lacking.

Response: We thank the review's kind suggestion. We have performed ESI-MS of the reaction mixture containing **R-1** and 2 eq. $\text{Zn}(\text{BF}_4)_2$ and 3 eq. potassium oxalate at different reaction time and showed as **Fig. R5** (Supplementary **Fig. 20**). $[(\text{L}^1)_2\text{Zn}_4(\text{ox})_2(\text{BF}_4)_2(\text{H}_2\text{O})_1]^{2+}$ is believed to be the intermediate when transforming **R-1** to **R-2-Zn**, which is identified by the ESI-MS. We have added "*ESI-MS of the transforming mixture suggested $[(\text{L}^1)_2\text{Zn}_4(\text{ox})_2(\text{BF}_4)_2(\text{H}_2\text{O})_1]^{2+}$ may be a potential intermediate (Supplementary **Fig. 20**).*" into the discussion of the transforming **R-1** to **R-2-Zn**.

Fig. R5 (Supplementary **Fig. 20**). a) ESI-MS spectra of reaction mixture containing *R-1* and 2 eq. $Zn(BF_4)_2$ and 3 eq. potassium oxalate at different reaction time with a key intermediate highlighted and a proposed structure showed, b) experimental and simulated isotopic patterns of intermediates $[(L^1)_2Zn_4(ox)_2(BF_4)_2(H_2O)_1]^{2+}$, c) experimental and simulated isotopic patterns of fragment $[(L^1)_2Zn_1(BF_4)]^{1+}$ from *R-1*. The $[(L^1)_2Zn_4(ox)_2(BF_4)_2(H_2O)_1]^{2+}$ is almost one-third of the final structure of *R-2-Zn*, and the intensity of this peak rises as time increases, while that of the fragment $[(L^1)_2Zn_1(BF_4)]^{1+}$ from *R-1* lowers down. $[(L^1)_2Zn_4(ox)_2(BF_4)_2(H_2O)_1]^{2+}$ is believed to be the intermediate when transforming *R-1* to *R-2-Zn*.

5. Fig. 4g, what is the condition for these UV measurements, solid state?

Response: We thank the review for pointing this out. The UV-Vis spectra were for solution samples. We have added "*samples were dissolved in acetonitrile;*" in the caption of Fig. 4g.

6. The sequence of supplementary figure in manuscript should rearrange in a better order. E.g. line 247, FigS.50, line 288-291, FigS43-48.

Response: We appreciate the reviewer's valuable suggestion. In response, we relocated the "Density Functional Theory (DFT) Calculations" section after the "Synthesis" section to enhance the organization of the supplementary figures. We also adjusted the order of some supplementary figures. However, division of the supplementary figures into multiple sections is necessary for improved clarity. Consequently, certain supplementary figures with higher numerical order may still precede those with smaller numbers in the text.

7. Fig 1a: it is better to add relative equivalent of the materials. It is difficult to realize whether excess of Zn²⁺ was added in the reaction mixture or just a suitable amount was added instead.

(the figures like this one at times are over cluttered – maybe the reaction schemes should be separated from the crystal structures – (the crystals can also have the hydrogens omitted for clarity) and also have some of the symmetry elements indicated.

Response: We thank the review's kind suggestion. We have added relative equivalent of the materials into **Fig. 1a**. We have also adjusted the space between **Fig. 1a** and the crystal structures to make it looks neat. The hydrogens of all structure in the figures have been omitted for clarity. Symmetry elements of *R-1* and *R-2-Cr* have been added in **Fig. 1b** and **1c**. New **Fig. 1**, **Fig. 2**, **Fig. 3** and **Fig. 4** have been provided in the revised manuscript.

8. [ox] is using in the paragraph but [C₂O₄] is using in the scheme. They should be consistent in the whole manuscript.

Response: We thank the review's kind suggestion. We have changed all the [C₂O₄] in the figures and text into [ox] to make sure the whole manuscript is consistent.

9. Line 118: it should be (b) R-1.

Response: We thank the review for pointing this out. We have revised accordingly and rechecked the whole manuscript to make sure no such mistakes anymore.

10. References list: for example in reference 1, it should be Bols, P. S., Anderson, H. L. etc

Response: We thank the review's kind suggestion. We have revised accordingly.

Reviewers' Comments:

Reviewer #1:

Remarks to the Author:

Basically, the authors properly responded to the reviewers comments and revised their manuscript. Thus, I would like to recommend the publication of this paper in Nature Communications.

Reviewer #2:

Remarks to the Author:

Many thanks to the authors who have responded to all the crystallographic points raised in my original review. I am happy to recommend publication after attending to a minor point described below:

I have a suggested correction about the handling of geometric restraints that the authors should consider. The authors have correctly removed the rigid hexagonal geometric restraints (AFIX 66) from the pyridine rings from the structures, however, in some cases (e.g. structure R-3) they have been replaced with inappropriately applied geometric similarity restraints (SADI) instead. For example in structure R-3 pyridyl ring C11-N16 has similarity restraints (SADI) applied to all the 1,2 bonds around the ring - both C-C bonds and C-N bonds. This restraint is applied incorrectly as given the asymmetric chemical environment of this ring these bonds would not be expected to have equal lengths - an accurately modelled pyridine ring is never a perfect hexagon. Instead, similarity restraints should be applied to chemically identical bonds in different pyridine rings in the structure e.g. "SADI N16 C11 N63 C64 N54 C49" and "SADI C11 C12 C49 C50 C64 C65" (the SAME command is a very efficient way of generating a large number of such restraints). This will allow plenty of geometrically correct restraints to be generated and won't push the pyridine rings towards and incorrect regular hexagonal geometry.

Although applying these changes will make the structure more chemically plausible it will make very little difference to the refinement statistics and not change any conclusions already drawn from the data, hence, there is no need for this work to be re-reviewed after the recommended changes have been made.

Reviewer #3:

Remarks to the Author:

The authors have satisfactorily addressed my previous suggestions for revision.

Response to Referees

Reviewer #1 (Remarks to the Author):

Basically, the authors properly responded to the reviewers comments and revised their manuscript.

Thus, I would like to recommend the publication of this paper in Nature Communications.

Response: We greatly appreciate the positive comments from the reviewer as well as the support of publishing this manuscript in *Nat. Commun.*

Reviewer #2 (Remarks to the Author):

Many thanks to the authors who have responded to all the crystallographic points raised in my original review. I am happy to recommend publication after attending to a minor point described below:

I have a suggested correction about the handling of geometric restraints that the authors should consider. The authors have correctly removed the rigid hexagonal geometric restraints (AFIX 66) from the pyridine rings from the structures, however, in some cases (e.g. structure R-3) they have been replaced with inappropriately applied geometric similarity restraints (SADI) instead. For example in structure R-3 pyridyl ring C11-N16 has similarity restraints (SADI) applied to all the 1,2 bonds around the ring - both C-C bonds and C-N bonds. This restraint is applied incorrectly as given the asymmetric chemical environment of this ring these bonds would not be expected to have equal lengths - an accurately modelled pyridine ring is never a perfect hexagon. Instead, similarity restraints should be applied to chemically identical bonds in different pyridine rings in the structure e.g. "SADI N16 C11 N63 C64 N54 C49" and "SADI C11 C12 C49 C50 C64 C65" (the SAME command is a very efficient way of generating a large number of such restraints). This will allow plenty of geometrically correct restraints to be generated and won't push the pyridine rings towards and incorrect regular hexagonal geometry.

Although applying these changes will make the structure more chemically plausible it will make very little difference to the refinement statistics and not change any conclusions already drawn from the data, hence, there is no need for this work to be re-reviewed after the recommended changes have been made.

Response: We thank the reviewer for the detailed comments that helped us clearly improve our results. We have revised all the 'SADI' restraints relating to the pyridyl rings for *R-1*, *R-2-Cr*, and

S-6-Zn-O₂ following the reviewer's instructions. The structures are now more reasonable. New .cif files have been provided.

Reviewer #3 (Remarks to the Author):

The authors have satisfactorily addressed my previous suggestions for revision.

Response: We greatly appreciate the positive comments from the reviewer as well as the support of publishing this manuscript in *Nat. Commun.*